# Reduced External Iliac Venous Blood Flow Rate Is Associated with Asymptomatic Compression of the Common Iliac Veins

**DOI:** 10.3390/medicina57080835

**Published:** 2021-08-18

**Authors:** Yuan-Hsi Tseng, Chien-Wei Chen, Min-Yi Wong, Teng-Yao Yang, Yu-Hui Lin, Bor-Shyh Lin, Yao-Kuang Huang

**Affiliations:** 1Division of Thoracic and Cardiovascular Surgery, Chiayi Chang Gung Memorial Hospital, College of Medicine, Chia-Yi and Chang Gung University, Taoyuan 33302, Taiwan; 8802003@cgmh.org.tw (Y.-H.T.); mynyy001@gmail.com (M.-Y.W.); vw200162@cgmh.org.tw (Y.-H.L.); 2Department of Diagnostic Radiology, Chiayi Chang Gung Memorial Hospital, College of Medicine, Chia-Yi and Chang Gung University, Taoyuan 33302, Taiwan; chienwei33@gmail.com; 3Institute of Medicine, Chung Shan Medical University, Taichung 40201, Taiwan; 4Institute of Imaging and Biomedical Photonics, National Chiao Tung University, Tainan 30010, Taiwan; borshyhlin@gmail.com; 5Department of Cardiology, Chiayi Chang Gung Memorial Hospital, College of Medicine, Chia-Yi and Chang Gung University, Taoyuan 33302, Taiwan; 2859@adm.cgmh.org.tw

**Keywords:** TRANCE MRI, non-contrast venography, QFlow, common iliac vein compression, May–Thurner syndrome

## Abstract

*Background and Objectives:* Compression of the common iliac veins (CIV) is not always associated with lower extremity symptoms. This study analyzed this issue from the perspective of patient venous blood flow changes using quantitative flow magnetic resonance imaging. *Materials and Methods:* After we excluded patients with active deep vein thrombosis, the mean flux (MF) and mean velocity (MV) of the popliteal vein, femoral vein, and external iliac vein (EIV) were compared between the left and right sides. *Results:* Overall, 26 of the patients had unilateral CIV compression, of which 16 patients had symptoms. No significant differences were noted in the MF or MV of the veins between the two sides. However, for the 10 patients without symptoms, the EIV MF of the compression side was significantly lower than the EIV MF of the non-compression side (*p* = 0.04). The receiver operating characteristic curve and chi-squared analyses showed that when the percentage difference of EIV MF between the compression and non-compression sides was ≤−18.5%, the relative risk of associated lower extremity symptoms was 0.44 (*p* = 0.016). *Conclusions:* If a person has compression of the CIV, a decrease in EIV blood flow rate on the compression side reduces the rate of symptom occurrence.

## 1. Introduction

Compression of the right, left, or bilateral common iliac vein (CIV) or the inferior vena cava (IVC) by a right common iliac artery or tortuous aorta, which also includes May–Thurner syndrome (MTS), can cause venous outflow tract stenosis or occlusion. This can result in symptoms related to lower extremity venous hypertension or insufficiency, such as pain, leg heaviness, edema, skin change, venous ulcer, or deep vein thrombosis (DVT) [1,2,3]. Some suggest that the proportion of patients with this anatomy is underestimated, because some patients do not have any lower extremity symptoms or have delayed occurrence of symptoms. For example, the age at which symptoms first occurred was >60 years in some patients [4,5]. The fact that some patients do not have symptoms or have delayed onset of symptoms has previously been explained from an anatomical perspective, including differences in the severity of iliac vein compression or the development of intimal hyperplasia. However, with the advent of the triggered angiography non-contrast enhanced magnetic resonance imaging (TRANCE MRI) plus the quantitative flow (QFlow) technique (Figure 1), more and more patients with compression of the CIV and IVC without symptoms have been identified. This technique also allows the mechanism from the prospective of blood flow to be explored.

In this study, we analyzed blood flow parameters in the lower extremity veins of patients with and without CIV or IVC compression and investigated the association between changes in venous blood flow and whether associated lower extremity symptoms would occur.

## 2. Materials and Methods

The protocol for this retrospective study was approved by the Institutional Review Board of Chang Gung Memorial Hospital (CGMF IRB No. 202001942B0; approved date: 13 November 2020). Because the patient data were retrospective and anonymity was maintained, the need for signed informed consent was waived for this study.

This study analyzed the data of 61 patients who underwent TRANCE MRI with QFlow acquisition for suspected lower extremity symptoms related to venous disease between October 2018 and July 2020. Given that DVT interferes with blood flow analysis, we excluded 10 patients who had active DVT. Baseline demographics and clinical characteristics, including sex, age, comorbidities, and clinical venous symptoms, were studied. The clinical venous symptoms included edema, skin changes, and ulcer.

Details of the parameter settings for our lower extremity TRANCE MRI were described in our previous study [6]. The QFlow data included the blood flow parameters of the popliteal vein (PV), femoral vein (FV), and external iliac vein (EIV). Although a QFlow technique can yield various types of blood flow data, after careful consideration, we only used the mean flux (MF) and mean velocity (MV) in this study. MF was used instead of blood flow rate because QFlow does not provide direct blood flow rate data, and the flux integral can determine the flow rate. Furthermore, flow velocity is related to static pressure according to Bernoulli’s principle.

Iliocaval territory compression is complex, and can include compression of the unilateral CIV, bilateral CIVs, or distal IVC, while “indentation” in the MRV can also be observed (Figure 2). According to the imaging findings and clinical status of each patient, the patients were classified into three groups: those without compression (*n* = 19), those with unilateral CIV compression (*n* = 26), and those with IVC or bilateral CIV compression (*n* = 6). The patients with unilateral CIV compression were then further classified into those with ipsilateral symptoms (*n* = 16) and those without ipsilateral symptoms (*n* = 10). (Figure 3). To avoid inherent differences in venous blood flow among different people, when we analyzed the blood flow parameters, the left and right lower extremities of the patients in each group were compared and the compression and non-compression sides were compared.

### Statistical Methods

For the presentation of baseline demographics and clinical characteristics, the continuous variables are presented as median and interquartile ranges, and the categorical variables are presented as numbers and percentages.

For the comparison of two groups, we used the Mann–Whitney U test to analyze continuous variables and the chi-squared test and Fisher’s exact test for categorical variables. Furthermore, receiver operating characteristic (ROC) curves were used to examine the discriminant capacity of the variables and to determine the cut-off values using the Youden index. The statistical analyses were conducted using SPSS for Windows (Version 17.0; SPSS, Inc., Chicago, IL, USA).

## 3. Results

The median age was 62 years (interquartile range, 48–74 years), and the majority of the patients were female (76.5%). Hypertension (33.3%), diabetes mellitus (21.6%), and hyperlipidemia (13.7%) were the major comorbidities. With regard to lower extremity symptoms, edema (35.3%) was the most common symptom, followed by ulcers (19.6%) and skin change (17.6%; Table 1).

To verify whether the venous blood parameter of the lower limbs was significantly different for the left and right sides in the absence of common iliac vein compression, the 19 patients without any CIV or distal IVC compression were analyzed through a comparison of their bilateral popliteal, femoral, and external iliac venous blood flow parameters (Table 2). As expected, no statistical differences were noted in the MF or MV of PVs, FVs, and EIVs of the lower extremities between the two sides. The MF and MV both rose sequentially from the PVs to FVs and EIVs, which was consistent with the normal physiology of lower limb venous blood flow.

Of the included patients, 26 had unilateral CIV compression. The influence of unilateral CIV compression on the blood flow was assessed by comparing the compression side and non-compression side lower extremities; however, this showed no statistically significant differences in the MF or MV of the PV, FV, or EIV. When patients in the “with ipsilateral symptoms” group (i.e., compression and lower extremity symptoms occurred in the same leg, *n* = 16) and “without ipsilateral symptoms” group (i.e., no lower extremity symptoms, or the compression and symptoms did not occur in the same leg, *n* = 10) were compared, the “without ipsilateral symptoms” group had a significantly lower EIV MF on the compression side (*p* = 0.04). Conversely, no statistical difference was noted in the EIV MF of the “with ipsilateral symptoms” group (Table 3).

To determine to what extent differences between the EIV MF of the left and right sides increase the likelihood of symptoms on the compression side, we calculated the percentage difference (PD) between the compression and non-compression sides as follows:PD = ((compression side value − noncompression side value)/(noncompression side value)) × 100%

ROC analysis was used to verify the discriminant capacity of the PDs and to identify cut-off values. The PD of EIV MF (*p* = 0.031) had statistical significance (Figure 4A). In addition, some patients had bilateral limb symptoms despite having unilateral EIV compression. Therefore, we excluded these patients (*n* = 4) and performed ROC curve analysis again; however, similar results were found (Figure 4B). Moreover, the cut-off values for both PDs of EIV MF were the same, at −18.50%. After the classification of patients using cut-off values, we revealed that the patients with PDs of EIV MF ≤ −18.50% had lower rates of occurrence of lower extremity symptoms than those with PDs of EIV MF > −18.50% (relative risk, 0.44; confidence interval, 0.22–0.94; *p* = 0.016).

Finally, for the patients (*n* = 6) who had bilateral CIV or distal IVC compression, it seems that the asymptomatic sides also have a trend toward lower EIV MF and MV (Figure 5). However, the bilateral differences in these parameters did not reach statistical significance because of the small sample size.

## 4. Discussion

Our study revealed counterintuitive results for compression-side EIV blood flux regarding the appearance of associated lower extremity symptoms. In this study, although >50% of the patients had unilateral or bilateral CIV compression, indicated by TRANCE MRI, their clinical manifestations were not always in line with imaging findings. Analysis of the QFlow data showed that asymptomatic CIV compression may be linked to whether the EIV blood flux in the compression side ever decreased beyond a certain level. With a sufficient decrease in the EIV blood flow rate of the compression limb, the probability of lower limb edema, skin change, and ulcer also decreased.

Possible explanations for this result include local blood flow regulation and the development of collateral vein formation or enlargement. However, the local mechanisms for the regulation of blood flow are usually present in the arteries and arterioles, with venous blood flow usually being influenced passively by changes in arterial blood flow [7,8,9]. However, we did not observe significant changes in the blood flux of the PVs or FVs. Therefore, the development of collateral vein formation or enlargement that keep up with the deterioration of the CIV stenosis is the more likely cause of asymptomatic CIV or IVC compression. Another possible reason is that when the pressure in the femoral vein is increased in common iliac vein obstruction, with the same resistance, it will increase the flow rate through the external iliac vein [10].

Asymptomatic CIV compression is not an uncommon phenomenon; Kibbe et al. (2004) reported that this anatomic variant existed in 24% of patients without symptoms [11]. The widely accepted explanation for asymptomatic CIV compression is that the probability of symptoms occurring correlates with the severity of iliocaval venous compression or the time required for spur formation [12]. The exacerbation of iliac venous compression due to progressive lumbar lordosis in women has also been described [13,14]. These theories provide good explanations for why some patients experience symptoms at a certain age, usually 20–50 years [15,16], but it is more difficult to explain why some people over 50 or 60 years old remain asymptomatic.

In our analysis, we excluded patients with ipsilateral DVT before calculating blood flow changes in the external iliac veins. The reasons for doing this was that ipsilateral DVT is undoubtedly an indicator of iliocaval venous compression syndrome if no other risk factors are evident. Historically, higher proportions of DVT in the left lower limb have resulted in the finding of MTS [17,18], and DVT has been viewed as the last stage of MTS [19]. In addition, DVT influences the accuracy of any blood flow parameter analysis. However, because edema, skin change, and ulcer are all related to venous hypertension or insufficiency, the data excluding the patients with DVT should still be representative of the change in EIV blood flow under CIV or IVC compression.

Aggressive intervention for iliofemoral venous thrombosis associated with MTS is beneficial for alleviating recurrent DVT or post-thrombotic syndrome [19,20]. Currently, endovascular interventions are the mainstream treatment options, including percutaneous transluminal angioplasty (PTA), stenting, and thrombolysis, and the safety and mid-term patency rates of these approaches have been confirmed [12,21,22]. At our hospital, two patients received PTA and stenting after complete DVT resolution, without any complications. However, at present, stenting for venous stenosis is not routinely performed at our hospital. The main concern regarding this approach is the occurrence of stent thrombosis [23,24]. Therefore, we believe that our study provides further information for determining the timing of endovascular surgery for the prevention of post-thrombotic syndrome.

For a patient with lower extremity symptoms related to venous disease, the usual noninvasive imaging modalities used are venous ultrasound, computed tomography venography (CTV), and magnetic resonance venography (MRV), whereas the commonly used invasive diagnostic modalities include intravascular ultrasound and angiography. However, these techniques have their limitations; venous duplex ultrasound cannot assess the abdominal and pelvic veins accurately and may erroneously lead a physician to ignore iliac vein or IVC compression [12,25]. CTV requires the use of a contrast medium and radiation exposure, whereas previously, MRV has been relatively time-consuming and expensive [1]. Moreover, the quality of an isolated MRV study is not sufficiently stable for evaluating the degree of compression [26]. Furthermore, venography plus intravascular ultrasound is invasive and requires the use of a contrast medium and radiation exposure.

Over the past 30 years, with the evolution of relevant techniques, MRV has yielded excellent imaging quality, even without gadolinium administration, making it suitable for patients who are allergic to the contrast medium or have impaired renal function. Moreover, with the advantage of no radiation exposure, it can be used on pregnant individuals. Furthermore, in terms of detecting DVT, MRV has sensitivity and specificity equal to, or higher than, those of Doppler sonography, especially for asymptomatic and obese patients [25,27,28]. TRANCE MRI, a novel technique, can clearly delineate the abdominal, pelvic, and lower limb arteries and veins [6,29]. In our hospital, some physicians have used TRANCE MRI as the first-line imaging modality for follow-ups to determine post-thrombotic changes.

The QFlow technique is a fully developed technology and has been used to study cerebral, cardiac, and aortic blood flow. In our hospital, we have used the QFlow technique to measure pelvic and lower limb venous blood flow. Although this application is rarely encountered, it should be applicable because, even though QFlow usually underestimates the blood flow velocity when blood flow is turbulent and when a vessel diameter is small [30,31,32,33,34], the iliac veins and FVs are usually large, and their blood flow usually presents as laminar flow; therefore, related blood flow measurements are generally accurate. Combining TRANCE MRI and QFlow provides a safe and convenient diagnostic tool for surveying and following up on lower limb venous disease. In addition, the interpretation of this imaging modality is less operator dependent than is the interpretation of duplex Doppler ultrasonography. Therefore, we believe that it is reasonable to recommend using this combined technique as the first-line imaging modality for evaluation of chronic venous disease.

Overtreatment is another important issue. Signs of severe CIV compression on the image examinations may lead the physicians to disregard the fact that the patient has no or only mild symptoms, meaning that treatment is unnecessary [35,36]. The situation may be even more serious if a patient were to undergo the examination in the supine position. In past studies, the blood flow of common iliac vein will further decrease as a result of the increase in the intensity of CIV compression when lying down, which may cause a misjudgment of the severity of compression [37]. Therefore, when severe CIV compression is highly suspected at TRANCE MRI images and QFlow data, other vascular examinations, such as intravascular ultrasound, should be considered to avoid posture-related artifacts.

### Limitations

As an evaluation of a new imaging technique, the major concern with the present study was the small sample size. Fortunately, we obtained a clear conclusion from a small amount of data. Although most of the patients had some degree of lower extremity venous symptoms, QFlow data still provided blood flow parameters with the normal or near-normal range of the side limb as a reference. Therefore, our results still have reference values for the evaluation of iliocaval territory compression.

The inability to reproduce the increase in blood flow rate caused by muscle contraction when walking or standing is a shortcoming of MRI and Qflow scanning due to the patient’s position at the time of examination. In future research, we may be able to roughly simulate the increase in blood flow rate resulting from muscle contraction through the use of compression stockings during the examination.

## 5. Conclusions

The results of this study demonstrate that if a person has compression of the CIV, a decrease in the ipsilateral EIV blood flow rate will reduce the probability of symptoms occurring related to venous hypertension or insufficiency. This result may help clinicians to decide whether to perform early treatment for iliocaval venous territory compression.

## Figures and Tables

**Figure 1 medicina-57-00835-f001:**
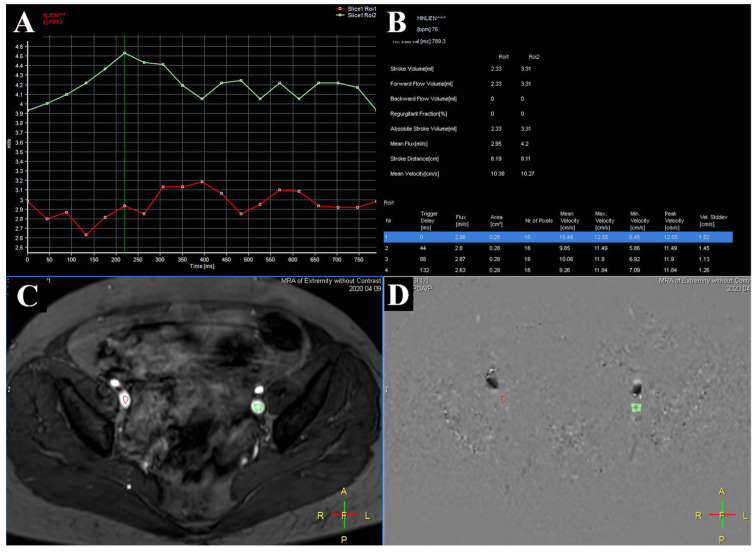
QFlow scanning through TRANCE-MRI. (**A**) Two external iliac veins with the flow sequence by time. (**B**) QFlow parameters with different trigger delays. (**C**) Areas of interest in both external iliac veins. (**D**) Image obtained during data acquisition.

**Figure 2 medicina-57-00835-f002:**
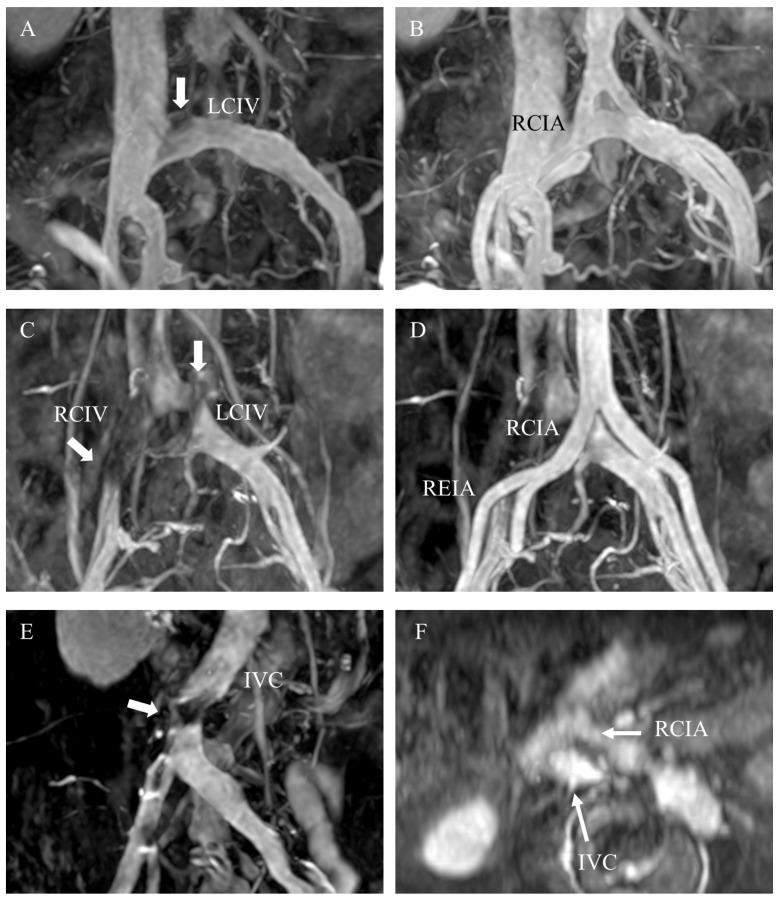
Examples of various venous indentations in triggered angiography non-contrast enhanced magnetic resonance imaging, including a left common iliac venous indentation (white arrow) (**A**) by the overlying right common iliac artery (**B**), bilateral common iliac venous indentations (white arrows) (**C**) by the right overlying right common and external iliac veins (**D**), and an inferior vena cava indentation (white arrow) (**E**) by the right common iliac artery (**F**). Abbreviations: IVC, inferior vena cava; LCIV, left common iliac vein; RCIA, right common iliac artery; RCIV, right common iliac vein; REIA, right external iliac artery.

**Figure 3 medicina-57-00835-f003:**
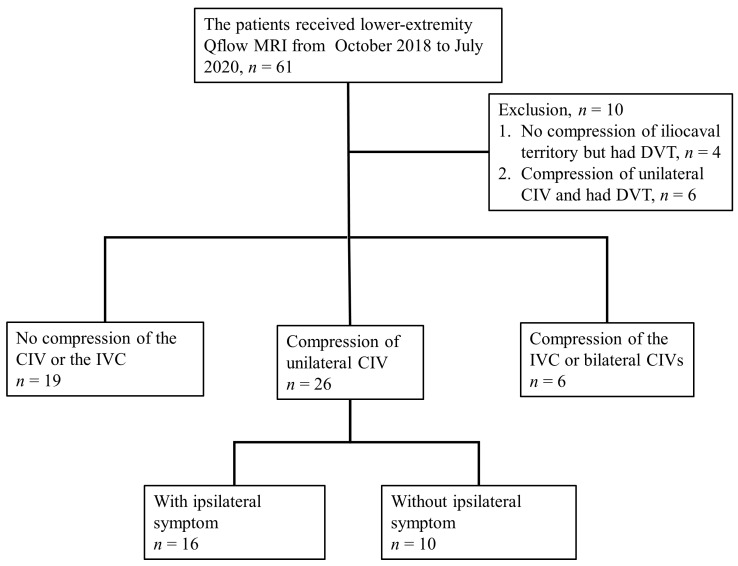
Hierarchy diagram. Abbreviations: CIV, common iliac vein; DVT, deep vein thrombosis; IVC, inferior vena cava; Qflow MRI, quantitative flow magnetic resonance imaging.

**Figure 4 medicina-57-00835-f004:**
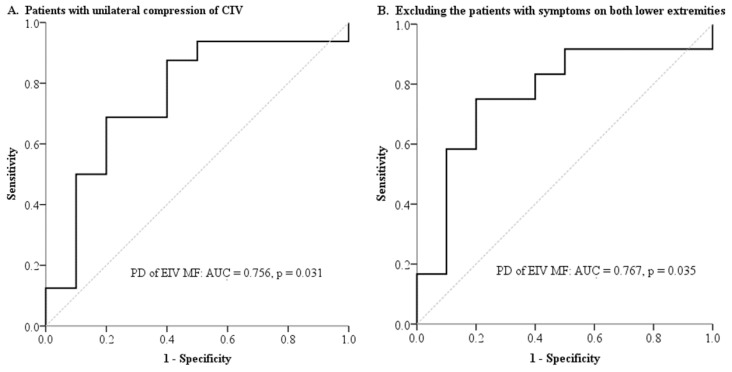
ROC curves for PD of EIV MF for predicting the occurrence of symptoms in the compression-side lower extremity (**A**) and after the exclusion of patients with bilateral symptoms (**B**). Abbreviations: DVT, deep vein thrombosis; EIV, external iliac vein; MF, mean flux; PD, percentage difference.

**Figure 5 medicina-57-00835-f005:**
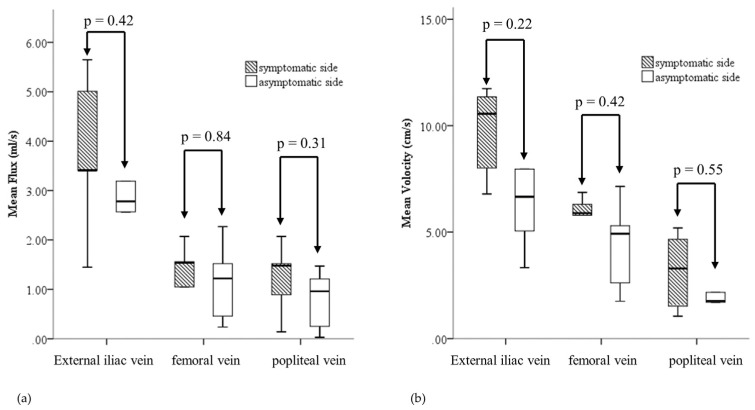
Boxplots comparing mean flux (**a**) and mean velocity (**b**) between the right and left popliteal veins, femoral veins, and external iliac veins of the patients with bilateral common iliac venous or inferior vena cava compression.

**Table 1 medicina-57-00835-t001:** Baseline demographics and clinical characteristics of all patients who underwent magnetic resonance imaging.

Variables	*N* = 51
Age (yrs)	62 (48, 74)
Sex, *n* (%)	
Male	12 (23.5)
Female	39 (76.5)
Comorbidities	
Hypertension, *n* (%)	17 (33.3)
Diabetes mellitus, *n* (%)	11 (21.6)
Hyperlipidemia, *n* (%)	7 (13.7)
History of cerebrovascular accident, *n* (%)	2 (3.9)
Chronic kidney disease, *n* (%)	3 (5.9)
Coronary artery disease, *n* (%)	2 (3.9)
Atrial fibrillation, *n* (%)	2 (3.9)
Smoking, *n* (%)	
No	44 (86.3)
Yes	6 (11.8)
Quit	1 (2.0)
Chronic lung disease, *n* (%)	2 (3.9)
Symptom and sign	
Edema, *n* (%)	18 (35.3)
Right, *n* (%)	7 (13.7)
Left, *n* (%)	8 (15.7)
Bilateral, *n* (%)	3 (5.9)
Skin changes, *n* (%)	9 (17.6)
Right, *n* (%)	7 (13.7)
Left, *n* (%)	1 (2.0)
Bilateral, *n* (%)	1 (2.0)
Ulcer, *n* (%)	10 (19.6)
Right, *n* (%)	4 (7.8)
Left, *n* (%)	6 (11.8)
Bilateral, *n* (%)	0 (0)

**Table 2 medicina-57-00835-t002:** Analysis of the bilateral lower extremities of patients without common iliac venous compression and deep vein thrombosis.

	Right Lower Extremity *N* = 19	Left Lower Extremity *N* = 19	*p*
EIV			
MF (mL/s)	2.56 (2.15, 5.33)	2.58 (1.56, 3.95)	0.26
MV (cm/s)	7.96 (4.26, 9.92)	6.51 (5.20, 10.63)	0.60
FV			
MF (mL/s)	1.00 (0.71, 1.61)	1.06 (0.57, 1.41)	0.73
MV (cm/s)	4.49 (2.97, 6.75)	3.56 (2.78, 5.62)	0.95
PV			
MF (mL/s)	0.56 (0.34, 1.08)	0.59 (0.35, 0.77)	0.75
MV (cm/s)	2.21 (1.38, 4.08)	2.27 (1.54, 3.09)	0.53

Data are presented as median [interquartile range]; EIV, external iliac vein; FV, femoral vein; MF, mean flux; MV, mean velocity; PV, popliteal vein.

**Table 3 medicina-57-00835-t003:** Analysis of bilateral lower extremities of the patients with unilateral common iliac venous compression but without deep vein thrombosis.

	With Ipsilateral Symptoms *N* = 16			Without Ipsilateral Symptoms *N* = 10		
	Non-Compression Side	Compression Side	*p*	Non-Compression Side	Compression Side	*p*
EIV						
MF (mL/s)	2.49 (1.72, 3.02)	2.37 (1.79, 4.58)	0.67	2.79 (1.77, 4.00)	1.67 (0.88, 1.90)	0.04
MV (cm/s)	7.38 (5.08, 9.92)	6.06 (5.30, 8.92)	0.56	6.65 (3.54, 9.00)	5.63 (2.84, 7.28)	0.32
FV						
MF (mL/s)	1.01 (0.72, 1.76)	1.15 (0.56, 2.13)	0.90	1.11 (0.95, 2.15)	1.00 (0.54, 1.25)	0.22
MV (cm/s)	4.34 (3.58, 6.71)	4.26 (2.74, 8.62)	0.93	5.12 (4.38, 6.86)	4.47 (3.67, 5.93)	0.28
PV						
MF (mL/s)	0.72 (0.48, 1.04)	0.68 (0.43, 1.22)	0.99	0.51 (0.22, 1.39)	0.52 (0.26, 0.68)	0.73
MV (cm/s)	2.26 (1.53, 3.87)	2.07 (1.45, 3.71)	0.90	1.91 (1.08, 3.19)	1.42 (1.03, 2.71)	0.49

Data are presented as median [interquartile range]; EIV, external iliac vein; FV, femoral vein; MF, mean flux; MV, mean velocity; PV, popliteal vein.

## Data Availability

Restrictions apply to the availability of these data. Data was obtained from Chiayi Chang Gung Memorial Hospital and are available from the authors with the permission of Chiayi Chang Gung Memorial Hospital.

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
