# Peer review of "Reduced External Iliac Venous Blood Flow Rate Is Associated with Asymptomatic Compression of the Common Iliac Veins"

_medicina, 2021, doi:10.3390/medicina57080835_

Round 1
Reviewer 1 Report
Very interesting results from a well performed study. Two major issues that need detailed discussion with appropriate references:
1.* The pressure in femoral vein is increased in CIV obstruction (J Thromb Haemost. 2016 Jun;14(6):1163-70. And other…). With the same resistance, it will increase the flow rate through the CIV – one of the explanations of your results.
2.** most importantly, symptoms, particularly venous claudication, are the result of the CIV inability to adjust (increase flow rate) in response to walking or prolong standing. Patient examination in supine position at rest will not address this factor.
Reviewer 2 Report
Asymptomatic Left Iliac Vein compression was described with veinography in 2018.
Your TRANCE MRI and QFlow and Veinography were both performed in supine position. .
Van Vuuren T.M.A.J.Kurstjens R.M.L.Wittens C.H.A.van Laanen J.H.H.de Graaf R.
Illusory angiographic signs of significant iliac vein compression in healthy volunteers.
Eur J Vasc Endovasc Surg. 2018; 56: 874-879
Your TRANCE MRI and QFlow and Veinography were both performed in supine position Both were performed in supine position.
We have shown with Duplex US total Left Iliac Vein total compression in supine position relieved with flow recovery in half-seated position.
Zamboni, P; Franceschi,C .Delfrate,R. The overtreatment of illusory May Thurner syndrome Paolo Veins and Lymphatics 2019; 8:8020]
Zamboni, P; Franceschi,C. How to Assess Illusory May-Thurner Syndrome by Ultrasound
Eur J Vasc Endovasc Surg 2019 Aug;58(2):305. doi: 10.1016/j.ejvs.2019.01.034
The live duplex scan flow stop and release is shown the Duplex US video
Pseudo MTS : https://www.youtube.com/watch?v=h931XXo2hdk&t=23s
This artifact could be cause of MTS overtreatments
So, the authors should add these studies to their article as discussion about the posture artefact.
Maybe, a half-seated MRI control on these people could be a relevant , especially to avoid useless stentings.
Round 2
Reviewer 2 Report
Improved article. good for publication